

# Data sharing policies of journals in life, health, and physical sciences indexed in Journal Citation Reports

Jihyun Kim[1], Soon Kim[2], Hye-Min Cho[3], Jae Hwa Chang[3] and Soo Young Kim[4]

[1] Department of Library and Information Science, Ewha Womans University, Seoul, South Korea
[2] Research Institute for Social Science, Ewha Womans University, Seoul, South Korea
[3] Infolumi Co, Seongnam, South Korea
[4] Department of Family Medicine, Kangdong Sacred Heart Hospital, Hallym University College of Medicine, Seoul, South Korea

## ABSTRACT

**Background**. Many scholarly journals have established their own data-related policies, which specify their enforcement of data sharing, the types of data to be submitted, and their procedures for making data available. However, except for the journal impact factor and the subject area, the factors associated with the overall strength of the data sharing policies of scholarly journals remain unknown. This study examines how factors, including impact factor, subject area, type of journal publisher, and geographical location of the publisher are related to the strength of the data sharing policy.

**Methods**. From each of the 178 categories of the Web of Science's 2017 edition of *Journal Citation Reports*, the top journals in each quartile (Q1, Q2, Q3, and Q4) were selected in December 2018. Of the resulting 709 journals (5%), 700 in the fields of life, health, and physical sciences were selected for analysis. Four of the authors independently reviewed the results of the journal website searches, categorized the journals' data sharing policies, and extracted the characteristics of individual journals. Univariable multinomial logistic regression analyses were initially conducted to determine whether there was a relationship between each factor and the strength of the data sharing policy. Based on the univariable analyses, a multivariable model was performed to further investigate the factors related to the presence and/or strength of the policy.

**Results**. Of the 700 journals, 308 (44.0%) had no data sharing policy, 125 (17.9%) had a weak policy, and 267 (38.1%) had a strong policy (expecting or mandating data sharing). The impact factor quartile was positively associated with the strength of the data sharing policies. Physical science journals were less likely to have a strong policy relative to a weak policy than Life science journals (relative risk ratio [RRR], 0.36; 95% CI [0.17–0.78]). Life science journals had a greater probability of having a weak policy relative to no policy than health science journals (RRR, 2.73; 95% CI [1.05–7.14]). Commercial publishers were more likely to have a weak policy relative to no policy than non-commercial publishers (RRR, 7.87; 95% CI, [3.98–15.57]). Journals by publishers in Europe, including the majority of those located in the United Kingdom and the Netherlands, were more likely to have a strong data sharing policy than a weak policy (RRR, 2.99; 95% CI [1.85–4.81]).

**Conclusions**. These findings may account for the increase in commercial publishers' engagement in data sharing and indicate that European national initiatives that

Corresponding author
Soo Young Kim,
hallymfm@gmail.com

encourage and mandate data sharing may influence the presence of a strong policy in the associated journals. Future research needs to explore the factors associated with varied degrees in the strength of a data sharing policy as well as more diverse characteristics of journals related to the policy strength.

## INTRODUCTION

The value of research data can be increased by making the data widely accessible to other researchers (i.e., data sharing) (*Mozersky et al., 2019*; *Wang & Lv, 2018*). Data sharing may improve the rigor and reproducibility of research, thereby contributing to scientific advancements (*Borgman, 2012*). Many scholarly journals are increasingly establishing their own data policies, which specify the level of enforcement of data sharing, the types of data to be submitted, and the procedures for making data available (*Gherghina & Katsanidou, 2013*).

Several studies have examined the factors associated with journals' adoption of data sharing policies. The impact factor of a journal is the most cited element linked to the strength of its data sharing policies. The strength ranges from no policy, to weak policy (unenforceable), to strong (enforceable) policy (*Piwowar & Chapman, 2008*). Some studies have indicated that journals with higher impact factors are more likely to require data sharing, as opposed to merely recommending it or not mentioning it at all (*Piwowar & Chapman, 2008*; *Sturges et al., 2015*; *Vasilevsky et al., 2017*). Another group of studies also identified a positive relationship between impact factor and whether or not journals had data sharing policies (*Aleixandre-Benavent et al., 2016*; *Crosas et al., 2019*; *Resnik et al., 2019*; *Stodden, Guo & Ma, 2013*; *Vidal-Infer et al., 2018*; *Zenk-Moltgen & Lepthien, 2014*). In other research, however, no relationship was found between impact factor and the adoption of data sharing policies (*Aleixandre-Benavent et al., 2019*; *Vidal-Infer et al., 2019*).

Several possible explanations for this inconsistency exist. One possibility is that previous studies were limited to specific areas (e.g., dentistry, *Vidal-Infer et al., 2018*; pediatrics, *Zenk-Moltgen & Lepthien, 2014*; and toxicology, *Aleixandre-Benavent et al., 2016*). A second possibility is the variations in the definition of data sharing policies across studies. A third possibility is that there might be a temporal trend; data sharing policies were initially established by journals with a higher impact factor and, over time, have been adopted by those with a lower impact factor.

In addition, disciplinary variations across journals influence the adoption of data sharing policies. One study analyzed the data sharing policies of 50 journals in the social sciences and demonstrated that the proportion of journals with data sharing policies varied across subject areas; for example, 74% of journals in economics had such policies, while only 18% of history journals did (*Crosas et al., 2019*). The results of another study suggested that biology journals were more likely to require data sharing than mathematics or social science journals (*Resnik et al., 2019*).

Additional factors besides impact factor and academic subject area were found to be significantly associated with the existence and strength of data sharing policies. The type of journal publisher was one such factor; as a previous study noted, non-commercial publishers, such as academic societies, were more likely to have a data sharing policy than commercial publishers (*Piwowar & Chapman, 2008*). A later study, which focused solely on for-profit publishers, suggested that journals published by Elsevier, Wiley, and scientific societies had a significantly higher probability of having data- or code-sharing policies than those published by other for-profit publishers (*Stodden, Guo & Ma, 2013*).

Geographical location is another factor that influences the strength of data sharing policies. Clinical trial data sharing policies, for example, have evolved differently in Europe compared to the United States, as the regulations governing data sharing practices vary by location. The European Union's Transparency Law made the proactive sharing of non-summary clinical trial data possible, but no similar act exists in the United States, where data sharing is reactive (*Westergren, 2016*).

In addition, data sharing requirements have been established primarily in the United States and several European countries, including the United Kingdom (UK) and the Netherlands. For example, the National Institutes of Health indicates they may expand upon existing data sharing requirements and require data management and sharing plans more broadly moving forward (*National Institute of Health, 2019*). UK Research and Innovation provides common principles on data policy, which emphasize making publicly funded research data openly available (*UK Research and Innovation, 2015*), while the Dutch government proposed the Netherland's Plan for Open Science to help make data easily accessible and reusable (*European Commission, 2018*). Furthermore, a larger European initiative called Plan S was proposed by Science Europe in 2018. Plan S requires all the results of publicly funded research in Europe to be published in open-access journals and/or deposited in open repositories with no embargoes. Plan S is another example that demonstrates that Europe has stronger open-access and open-data initiatives than other continents (*European Science Foundation, 2018*; *Kim, 2019*).

Minimal research has been conducted on the association between these journal aspects and the strength of data sharing policies across diverse fields of science. The current study thus examined whether the strength of the data sharing policies of journals in the fields of life, health, and physical sciences are associated with impact factor, subject area, type of publisher, and geographical location of the publisher. Exploring these factors will help elucidate which journal characteristics are related to the presence and/or strength of journal policies regarding data sharing.

## MATERIALS AND METHODS

### Data collection

In December 2018, the highest-ranked journal in each journal impact factor quartile (Q1, $0 < Z \leq 0.25$; Q2, $0.25 < Z \leq 0.5$; Q3, $0.5 < Z \leq 0.75$; or Q4, $0.75 < Z$; Z, percentile rank) was selected from each of the 178 categories of the 2017 edition of *Journal Citation Reports* (JCR). For example, *Physics of Life Reviews*, *Biochimica et Biophysica Acta-Biomembranes*,

**Table 1  Strength of data sharing policies.**

| Code | Strength of policy | Data sharing policy categories | Data availability statement is published | Data have been shared | Data have been peer-reviewed |
|---|---|---|---|---|---|
| 0 | No policy | No data sharing policy | No mention of data sharing | | |
| 1 | Weak policy | Encourages data sharing | Optional | Optional | Optional |
| 2 | | Expects data sharing | Required | Optional | Optional |
| 3 | Strong policy | Mandates data sharing | Required | Required | Optional |
| 4 | | Mandates data sharing and peer reviews data | Required | Required | Required |

Notes.

Modified from Wiley's policy for data sharing (https://authorservices.wiley.com/author-resources/Journal-Authors/open-access/data-sharing-citation/data-sharing-policy.html).

*Biointerphases*, and *Radiation and Environmental Biophysics* were selected for biophysics because they ranked first, respectively, among the Q1, Q2, Q3, and Q4 journals in biophysics. Because only one journal existed in the transportation category, we selected only this journal. From the remaining 177 categories, 708 journals were selected, comprising four journals (one in each of the four quartiles) in each of the 177 categories (i.e., 177 multiplied by 4). The selection process resulted in the original identification of 709 journals.

We then identified the journal subject areas based on four broad clusters, as defined by Scopus: life, health, physical, and social sciences (*Elsevier, 2020*). Although we selected the journals listed in JCR, the four subject areas were useful for creating a categorical variable that represented the subject areas. Because we focused on journals in life, health, and the physical sciences, we excluded five journals that were categorized solely under social sciences. In addition, we excluded four journals that were not listed in Scopus. As a result, the final set comprised 700 journals.

Each journal was assigned to one of the three subject areas—life, health, or physical sciences—or two or three of the subject areas simultaneously. According to Table 1, we identified 80 in life science, 154 in health science, and 305 in physical science. In terms of "multidiscipline" journals, we initially determined that 156 journals were assigned to more than one subject area. In addition, Scopus further divided the four subject areas into 26 subject categories plus one "general" category that contained multidisciplinary journals (*García, Rodriguez-Sánchez & Fdez-Valdivia, 2011*). We found five additional journals, including *Nature*, categorized as "general" without assigning broad subject clusters. We considered the 156 journals as being assigned to more than one broad subject area and the five journals in the general category as being multidisciplinary. Therefore, we included 161 journals in the "multidiscipline" category. The subject area variable therefore consisted of four categories: life science, health science, physical science, and multidiscipline. Moreover, of the 26 Scopus subcategories, we used 22 subject categories—all except those related to the social sciences and humanities—to perform further descriptive analyses of the relationship between subject areas and the strength of data sharing policies.

In addition, we determined the strength of the data sharing policies provided by the 700 journals based on a modified version of Wiley's data sharing policy (Table 2) (*Wiley, 2019*). This modification resulted from adding a "no policy" category to Wiley's policy and

**Table 2   General characteristics of the journals ($n = 700$).**

| Characteristics | | No. (%) | |
|---|---|---|---|
| Data sharing policy category | | | |
| No policy | No data sharing policy | 308 (44.0) | |
| Weak policy | Encourage data sharing | 125 (17.9) | |
| Strong policy | Expect data sharing | 267(38.1) | 170 (24.3) |
| | Mandate data sharing | | 71 (10.1) |
| | Mandate data sharing with peer review | | 26 (3.7) |
| Impact factor quartile[a] | | | |
| Q1 | | 176 (25.1) | |
| Q2 | | 174 (24.9) | |
| Q3 | | 176 (25.1) | |
| Q4 | | 174 (24.9) | |
| Type of publisher | | | |
| Commercial | | 531 (75.9) | |
| Non-commercial | | 169 (24.1) | |
| Location of journal publisher | | | |
| North America | | 318 (45.4) | |
| Europe | | 334 (47.7) | |
| Others | | 48 (6.9) | |
| Subject area | | | |
| Life science | | 80 (11.4) | |
| Health science | | 154 (22.0) | |
| Physical science | | 305 (43.6) | |
| Multidiscipline | | 161 (23.0) | |

**Notes.**

[a] Q1, first quartile; Q2, second quartile; Q3, third quartile; Q4, fourth quartile.

categorizing the strength of the policy into three levels—no, weak, and strong—according to *Piwowar & Chapman (2008)*.

The four authors of this study were separated into two pairs, each of which was assigned half of the journals. Each pair searched the assigned journals' web pages and information about their data sharing policies, using keywords such as "data sharing," "data availability," and "deposit." If there was no mention of data sharing on a journal's web page, the authors identified it as having no data sharing policy. They determined that a journal had a weak data sharing policy if it encouraged data sharing without any indication that it was mandated. As Table 2 illustrates, a strong data sharing policy included three degrees of strength: (1) expects data sharing, which requires publishing a data availability statement only; (2) mandates data sharing, which requires both a data availability statement and data sharing; or (3) mandates the sharing and peer review of data, which include all three requirements. The authors judged these criteria based on the inclusion of words that indicated obligation, such as "must," "should," or "require."

Within each group, two authors independently assigned relevant codes to the journals. The kappa statistic for measuring inter-rater reliability was 0.75, which indicated substantial

agreement (*McHugh, 2012*). After the initial coding, we compared the coding results and discussed the disparities to achieve agreement. The coding discrepancies were also discussed across the groups; accordingly, the four authors reached a consensus on the coding results.

The remaining two factors—type and geographical location of publisher—were also categorical variables. We presented type of publisher as a dichotomous variable—that is, commercial and non-commercial—as suggested by *Piwowar & Chapman (2008)*. The commercial publishers consisted of major publishers, such as Elsevier, Springer Nature, and Wiley, as well as a number of minor publishers. The non-commercial publishers were mostly academic societies or associations but also included several research institutes, university presses, and nonprofit foundations or organizations.

The geographical locations of the publishers were represented as continents, including North America, Europe, and others. With the exception of five in Canada, the North American publishers were located in the United States. The European publishers were located across 21 countries. The majority of the publishers comprised 178 from the UK and 60 from the Netherlands. Other publishers were located in Asia and Oceania.

## Statistical analysis

The data were analyzed using Stata ver. 16.1 (StataCorp., College Station, TX, USA). The strength of data sharing policies was a nominal variable with three categories: strong data sharing policy (hereafter, "strong policy"), weak data sharing policy (hereafter, "weak policy"), and no data sharing policy (hereafter, "no policy") (Table 2).

Both a univariable and a multivariable multinomial logistic regression analysis were conducted. Although the dependent variable had ordinal characteristics, we used multinomial rather than ordinal logistic regression because a test of the proportional odds assumption using ordinal regression demonstrated that the model failed to satisfy proportionality ($\chi^2(7) = 14.68$, $p = 0.04$). Since multinomial regression does not assume proportional odds, this was selected as the appropriate model.

Univariable multinomial logistic regression analyses were conducted to test the association of each factor with the strength of data sharing policies. All significant factors with $p < 0.05$ were included in the multivariable analysis. We assessed two-way interactions among the factor categories, and there was no interaction effect except between publisher location and subject area. To select best goodness of fit, we compared the Bayesian Information Criteria (BIC) of the main effect model and with interaction variables included; the main effect model had a lower BIC indicating better fit. To reduce the risk of overfitting, events per variable (EPV) was also calculated based on the smallest number of observations in the outcome categories divided by the number of effective regression coefficients (*De Jong et al., 2019*). Multivariable multinomial logistic regression analyses were then performed to determine whether the strength of data sharing policies can be predicted from the aforementioned factors.

Based on both of the analyses, we identified coefficients and standard errors and calculated the associated relative risk ratios (RRRs) and 95% confidence intervals (CIs). Wald tests for independent variables were also conducted to determine the significance of each independent variable on all pair-wise comparisons for the dependent variable.

## RESULTS

### Journal characteristics

Out of 14,223 journals in the JCR in December 2018, 709 (5%) were originally selected, and 700 in the life, health, and physical science categories were ultimately analyzed. Table 1 summarizes the general characteristics of the journals. Among the 700 journals, 308 (44.0%) had no data sharing policy, 125 (17.9%) had weak data sharing policies, and 267 (38.1%) had strong data sharing policies (at minimum expected data sharing).

Concerning the subject area variable, the journals that were categorized as "multidiscipline" included 156 that were classified under two or three subject areas, and five journals that fell under Scopus's "general" category. Specifically, there were 73 life and health science journals, 61 life and physical science journals, 14 health and physical science journals, 8 in all three areas, and five journals in the general category. Thus, 307 journals—including 80 in life sciences, 154 in health sciences, and 73 in both—were particularly relevant to the life and/or health sciences, although almost all the journals in the "multidiscipline" category were somehow related to either of these two subject areas.

The most common location for publishers was Europe (334, 47.7%), followed by North America (318 journals, 45.4%), which mostly comprised the United States. The number of commercial publishers (531, 75.9%) was more than three times that of non-commercial publishers. The top three commercial publishers were Elsevier (147 journals), Springer Nature (118 journals), and Wiley (72 journals).

### Univariable analysis of factors associated with the strength of data sharing policies

To examine the associations between the strength of data sharing policies and each factor, we carried out univariable multinomial logistic regression analyses. "Weak policy" was set as a base outcome category. Among the correlates that were tested individually to determine their relationships to policy strength, all factors showed significant associations ($p < 0.05$). Comparing no policy to weak policy, type of publisher was significant. Impact factor quartile, publisher location, and subject area were also significant when comparing strong to weak policy (Table 3).

Wald tests for independent variables in the univariable analysis were also performed to determine the significance of an independent variable across all pair-wise comparisons of the category levels (no, weak, and strong policy). Table 3 showed that overall, all independent variables were significant at the $p < 0.001$ level. Therefore, we rejected the null hypothesis that the coefficients of the dummy variables indicating each independent variable were jointly equal to zero.

Based on the univariable analysis, all the factors were eligible for inclusion in the multivariable model. All possible two-way interactions for the variables selected for the multivariable model were also tested but were not statistically significant at the $p = 0.05$ level, with the exception of the interaction variable between publisher location (Europe) and subject area (health science and physical science). We compared the model fit between the main-effect model that included the four factors only and the model with the interaction variable. As seen in Table 4, the addition of the interaction variable did not improve the

**Table 3   Results of univariable multinomial logistic regression analysis and wald tests for independent variables.**

| Statistics | No policy vs. Weak policy | | | Strong policy vs. Weak policy | | | Wald tests |
|---|---|---|---|---|---|---|---|
| Factors | RRR[a] | 95% CI | *p*-value | RRR | 95% CI | *p*-value | *p*-value |
| Impact factor quartile[b] | | | | | | | <0.001 |
| Q1 | 1(Ref) | – | – | 1(Ref) | – | – | – |
| Q2 | 0.63 | 0.34–1.19 | 0.15 | 0.72 | 0.40–1.30 | 0.28 | 0.36 |
| Q3 | 0.82 | 0.44–1.50 | 0.51 | **0.52** | **0.29–0.95** | **0.03** | 0.05 |
| Q4 | 1.55 | 0.84–2.86 | 0.16 | **0.26** | **0.13–0.51** | **<0.001** | <0.001 |
| Type of publisher | | | | | | | |
| Commercial | 1(Ref) | – | – | 1(Ref) | – | – | – |
| Non-commercial | **7.08** | **3.66–13.69** | **<0.001** | 1.46 | 0.71–3.00 | 0.30 | **<0.001** |
| Location of journal publisher | | | | | | | <0.001 |
| North America | 1(Ref) | – | – | 1(Ref) | – | – | – |
| Europe | 0.79 | 0.51–1.21 | 0.28 | **2.17** | **1.40–3.38** | <0.001 | <0.001 |
| Others | 2.15 | 0.86–5.37 | 0.10 | 1.07 | 0.36–3.15 | 0.09 | 0.09 |
| Subject area | | | | | | | <0.001 |
| Life science | 1(Ref) | – | – | 1(Ref) | – | – | – |
| Health science | 2.54 | 1.01–6.40 | 0.05 | 1.59 | 0.6-2–4.06 | 0.33 | 0.08 |
| Physical science | 0.50 | 0.24–1.03 | 0.06 | **0.38** | **0.18–0.80** | **0.01** | 0.04 |
| Multi-discipline | 0.80 | 0.35–1.85 | 0.60 | 1.22 | 0.53–2.79 | 0.63 | 0.36 |

**Notes.**
[a] RRR, relative risk ratio.
[b] Q1, first quartile, Q2, second quartile, Q3, third quartile, Q4, fourth quartile.

**Table 4   Comparison of the model fit.**

| Fit statistics / Model | | Main-effect model | Model with the interaction variable |
|---|---|---|---|
| Log likelihood ratio tests | Chi-square | 234.17 | 273.71 |
| | df | 18 | 30 |
| | *p*-value | <0.001 | <0.001 |
| BIC[a] | | 1347.96 | 1387.03 |

**Notes.**
[a] Bayesian information criteria.

model fit. The Bayesian information criteria (BIC) was smaller in the main-effect model, and the difference in BICs was 39.07; this is much greater than 10, which is strong evidence for choosing the main-effect model (*Raftery, 1995*). Therefore, our final model for the multivariable analysis consisted of the four factors without interaction terms. EPV was 6.94 (125 divided by 18), which exceeded the minimum EPV values of between 5 and 20 for reliable results (*Ogundimu, Altman & Collins, 2016*).

## Multivariable analysis of the factors associated with the strength of data sharing policies

Multivariable multinomial logistic regression analyses were performed to determine whether the four journal characteristics—impact factor, type of publisher, location of

**Table 5  Results of multivariable multinomial logistic regression analysis and wald tests for independent variables.**

| Statistics factors | No policy vs. Weak policy | | | Strong policy vs. Weak policy | | | Wald tests |
|---|---|---|---|---|---|---|---|
| | RRR[a] | 95% CI | p-value | RRR | 95% CI | p-value | p-value |
| Impact factor quartile[b] | | | | | | | <0.001 |
| Q1 | 1(Ref) | – | – | 1(Ref) | – | – | – |
| Q2 | 0.70 | 0.36–1.38 | 0.31 | 0.70 | 0.37–1.30 | 0.26 | 0.50 |
| Q3 | 0.95 | 0.50–1.84 | 0.89 | **0.51** | **0.27–0.96** | **0.04** | 0.02 |
| Q4 | 1.70 | 0.87–3.29 | 0.12 | **0.24** | **0.12–0.49** | **<0.001** | <0.001 |
| Type of publisher | | | | | | | |
| Commercial | 1(Ref) | – | – | 1(Ref) | – | – | – |
| Non-commercial | **7.87** | **3.98–15.57** | **<0.001** | 1.87 | 0.89–3.94 | 0.10 | **<0.001** |
| Location of journal publisher | | | | | | | **<0.001** |
| North America | 1(Ref) | – | – | 1(Ref) | – | – | – |
| Europe | 1.13 | 0.70–1.81 | 0.62 | **2.99** | **1.85–4.81** | **<0.001** | **<0.001** |
| Others | 1.39 | 0.51–3.79 | 0.52 | 1.33 | 0.42–4.15 | 0.81 | 0.81 |
| Subject area | | | | | | | **<0.001** |
| Life science | 1(Ref) | – | – | 1(Ref) | – | – | – |
| Health science | **2.73** | **1.05–7.14** | **0.04** | 1.98 | 0.75–5.19 | 0.17 | 0.12 |
| Physical science | 0.46 | 0.21–1.01 | 0.05 | **0.36** | **0.17–0.78** | **0.01** | 0.03 |
| Multi-discipline | 0.80 | 0.33–1.92 | 0.61 | 1.37 | 0.58–3.23 | 0.47 | 0.27 |

**Notes.**
[a]RRR, relative risk ratio.
[b]Q1, first quartile; Q2, second quartile; Q3, third quartile; Q4, fourth quartile.

publisher, and subject area—predict the strength of data sharing policies. The reference group of the outcome variable was "weak policy." Overall, the multinomial logistic regression model was significant ($\chi^2(18) = 234.17$, $p < 0.001$).

According to Table 5, the multinomial log-odds for having no policy relative to a weak policy was related to type of publisher and subject area. Specifically, journals from non-commercial publishers (RRR, 7.87; 95% CI [3.98–15.57]) had a significantly higher likelihood of having no data sharing policy than those from commercial publishers. In addition, health science journals (RRR, 2.73; 95% CI [1.05–7.14]) had a significantly higher probability of having no data sharing policy than life science journals.

The multinomial log-odds for adopting a strong versus a weak policy were significantly associated with impact factor, location of publisher, and subject area. In particular, journals in impact factors Q3 (RRR, 0.51; 95% CI [0.27–0.96]) and Q4 (RRR, 0.24; 95% CI [0.12–0.49]) had a significantly lower likelihood of adopting a strong data sharing policy than those in Q1. Journals with publishers in Europe had a higher likelihood of adopting a strong policy (RRR, 2.99; 95% CI [1.85–4.81]) than those with publishers in North America. Physical science journals (RRR, 0.36; 95% CI [0.17–0.78]) had a significantly lower probability of having a strong data sharing policy than life science journals.

In particular, we plotted the predicted probabilities of the impact factor quartiles for the strength of the data sharing policies to examine the patterns of policy strength by the journals' impact factor quartiles. Figure 1 shows that the probability of a journal having

no data sharing policy increases as its impact factor decreases. In contrast, the probability of a journal having a strong data sharing policy increases as its impact factor increases. No linear trend for journals with weak policies was identified with regard to their impact factor quartile rankings.

We also performed Wald tests for independent variables, which suggested that overall, all factors were significant at the $p < 0.001$ level (Table 5). We could reject the null hypothesis that the coefficients for the dummy variables indicating each factor were simultaneously equal to zero. The results suggested that all the factors should be included in the multivariable model.

## Distribution of policy strength across Scopus subject categories

The multinomial logistic regression analyses indicated that compared to the life science journals, the health science journals were more likely to have no policy relative to a weak policy and that the physical science journals had a lower likelihood of having a strong policy relative to a weak policy. The results implied that the life science journals had a greater probability of having strong data sharing policies than the physical science journals did. The life science journals were also more likely to have at least weak data sharing policies than the health science journals.

The findings were consistent with the analysis of subject areas based on the proportion of journals having strong data sharing policies across 22 Scopus subcategories of subject. According to Table 6, neuroscience (68%) had the highest proportion of journals with a strong policy, followed by immunology and microbiology (65%); environmental science (53%); biochemistry, genetics, and molecular biology (51%); and chemical engineering (46%). With the exception of journals in chemical engineering, subject categories with a high proportion of journals with strong data sharing policies were closely related to the life sciences.

The category with the lowest proportion of journals with a strong policy was mathematics (11%), followed by computer science (28%), health professions (29%), and nursing and veterinary journals (33%) (Table 6). Mathematics and computer science were considered fields in the physical sciences, and the remaining subjects were categorized as health sciences. Journals in these fields tended not to have strong data sharing policies.

## DISCUSSION

The results of the multinomial logistic regression analyses presented in Tables 3 and 5 demonstrate generally consistent findings in both the univariable (unadjusted) and multivariable (adjusted) models. In comparing journals with no data sharing policy relative to a weak policy, type of publishers was significant in both models and the effect size was higher in the multivariable model (unadjusted RRR, 7.08; adjusted RRR, 7.87). Subject area was not significant in the univariable analysis, but it was found to have a significant association with the strength of the data sharing policy in the multivariable model (adjusted RRR, 2.73). In comparing journals with a strong data sharing policy relative to a weak policy, impact factor quartile, journal publisher's location, and subject

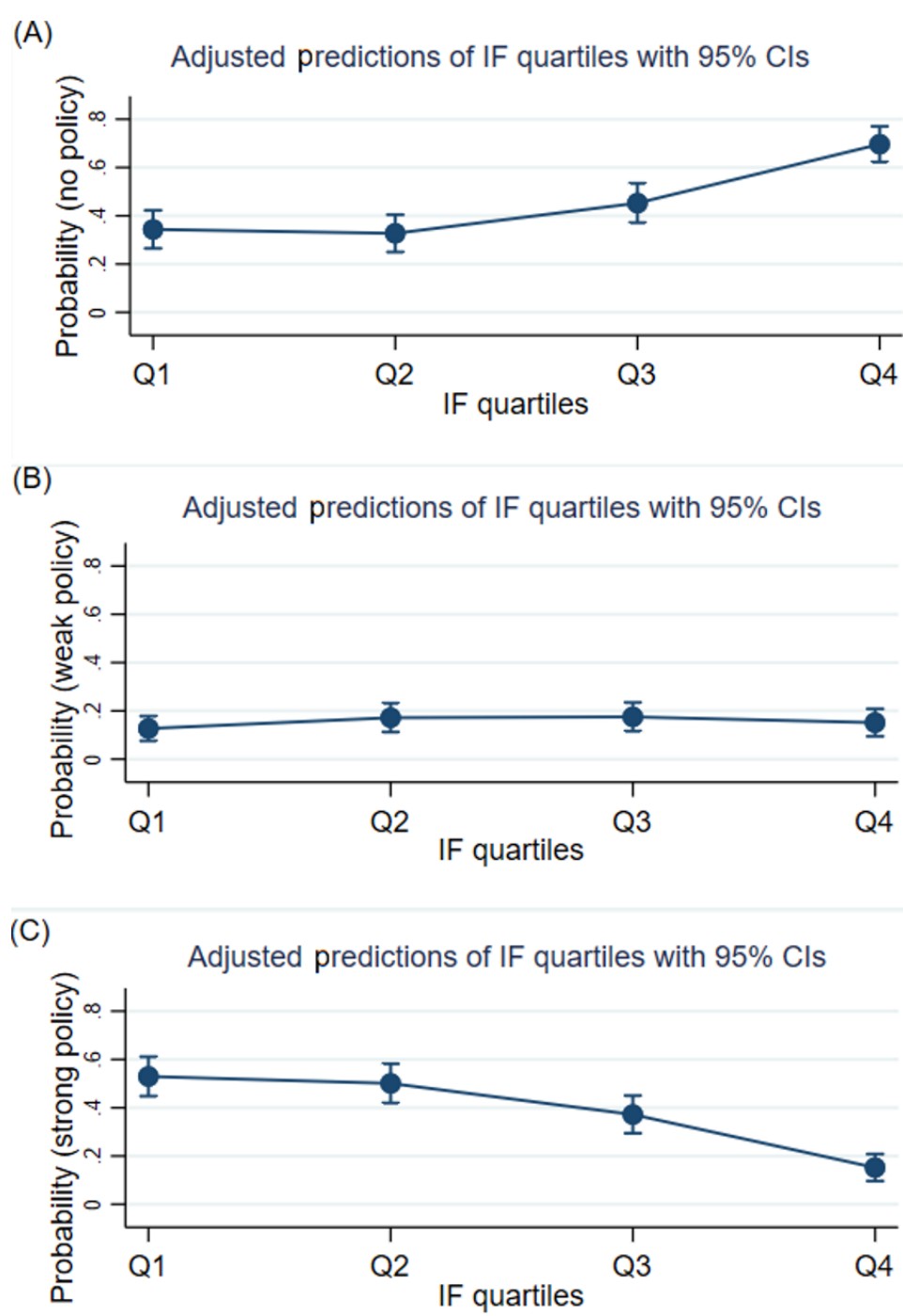

**Figure 1  Adjusted predictions of impact factor quartiles for the strength of data sharing policies.** (A) Adjusted prediction of impact factor quartiles for "no policy" category, (B) adjusted prediction of impact factor quartiles for "weak policy" category, and (C) adjusted prediction of impact factor quartiles for "strong policy" category. IF, impact factor; CI, confidence interval.

**Table 6  The rankings of top and bottom five categories according to the number of journals with strong policies.**

| Ranking | Top 5 | Bottom 5 |
|---|---|---|
| 1 | Neuroscience (68%) | Mathematics (11%) |
| 2 | Immunology and microbiology (65%) | Computer science (28%) |
| 3 | Environmental science (53%) | Health professions (29%) |
| 4 | Biochemistry, genetics, and molecular biology (51%) | Nursing (33%) |
| 5 | Chemical engineering (46%) | Veterinary (33%) |

area were significant in both models. In particular, the effect size of the publishers' location was greater in the multivariable model (unadjusted RRR, 2.17; adjusted RRR, 2.99).

To summarize the overall results, approximately one-third of the 700 JCR journals had a strong data sharing policy (at a minimum, they expected data sharing). The factors relating to whether journals had no data sharing policy or a weak policy included type of publisher and subject area, when controlling for other variables. Journal impact factor, geographical location of the publisher, and subject area were significantly associated with whether journals had a strong policy or a weak policy, when other variables were held constant.

In this study, within the strong policy category, 38.1% of the journals at least expected data sharing, while 13.8% mandated data sharing. While it is difficult to make accurate comparisons because of the use of different standards, the percentage of journals that mandated data sharing seems to be somewhat higher than reported in previous studies. *Vasilevsky et al. (2017)* reported that 11.9% of the journals in the 2013 JCR required data sharing for publications, and a survey of the editors of scientific journals in Korea found that only 3.4% of journals mandated data sharing (*Kim, Yi & Huh, 2019*). When a data sharing policy was defined as including statements regarding complementary material, significant differences in the percentages of the journals requiring were found among different domains: 83.5% for pediatric journals (*Aleixandre-Benavent et al., 2019*); 50% for information science and library science journals (*Sturges et al., 2015*); 47% for dental journals (*Vidal-Infer et al., 2018*); and 4.7% for scientific journals on substance abuse (*Resnik et al., 2019*).

Journal impact factor did not appear to have a significant association with whether journals had no policy or a weak policy, although several studies suggested that impact factor was positively related to having data sharing policies (*Aleixandre-Benavent et al., 2016*; *Crosas et al., 2019*; *Resnik et al., 2019*; *Stodden, Guo & Ma, 2013*; *Vidal-Infer et al., 2018*; *Zenk-Moltgen & Lepthien, 2014*). However, we did identify a significant association between impact factor and having a strong data sharing policy relative to a weak policy. Similar results were reported by previous studies (*Piwowar & Chapman, 2008*; *Sturges et al., 2015*; *Vasilevsky et al., 2017*). Particularly, we found that journals in Q1 were more likely to have a strong data sharing policy than a weak policy, in comparison to journals in Q3 and Q4. Overall, the findings were partially consistent with the results reported in existing research.

Our findings demonstrated that the subject areas of the journals predicted the strength of the data sharing policies: life science journals were more likely to have a strong policy relative to a weak policy than physical science journals, and they tended to have a weak policy relative to no policy in comparison to health science journals. The descriptive analysis also showed that the rate of adoption of a strong data sharing policy was higher for journals in the field of life sciences, such as neuroscience (68%) and immunology and microbiology (65%), than journals in the fields of physical sciences, such as mathematics (11%) and computer science (28%). The results were consistent with those reported by *Resnik et al. (2019)*, suggesting that journals in the field of biology tend to mandate data sharing more than those in the fields of mathematics or social sciences. Similarly, *Hrynaszkiewicz et al. (2017)* reported that life science journals tended to have weak or strong data sharing policies. Data sharing practices also differed by discipline due to variations in the researchers' attitudes toward data management and data sharing, as well as the infrastructure and expertise of data curation (*Key Perspectives, 2010*). Existing data sharing norms and practices in the field of life sciences would contribute to establishing these data sharing policies (*Pham-Kanter, Zinner & Campbell, 2014*).

Moreover, the type of publisher was associated with whether journals have no data sharing policy relative to a weak policy. That is, journals by commercial publishers were more likely to have a weak policy rather than no policy in comparison to non-commercial publishers. This finding is in opposition to that of *Piwowar & Chapman (2008)*, who found that non-commercial publishers were more likely to have data sharing policies because they supported data sharing more promptly than commercial publishers. Their study was published in 2008, and we considered the inconsistency as being reflective of changes in the commercial publishers' attitudes toward data sharing since that time, particularly those of major commercial publishers. According to *Stodden, Guo & Ma (2013)*, major publishers, such as Elsevier and Wiley, are more likely to have data sharing policies than other for-profit publishers.

The geographical location of publishers was a significant factor associated with the strength of the journals' data sharing policies. Specifically, journals from publishers in Europe were more likely to have a strong policy relative to a weak policy than journals from publishers in North America. However, there was no significant association between the location of the publishers and whether the journals have no policy versus a weak policy. These findings imply that journals from publishers in Europe are likely to have strong data sharing policies since the presence of national initiatives, such as Plan S in Europe, might influence the strength of the data sharing policies of the journals in those locations.

Scholarly journals are one of the major formal scientific communication channels that pressure authors to engage in data sharing, and those data sharing requirements have been found to have an impact on data sharing norms and behaviors (*Kim & Burns, 2016*). In this sense, journals play a significant role in fostering the culture of open science by establishing and implementing a data sharing policy. This study contributes to better understanding the interplay between journals' characteristics and the strength of data sharing policies. It also addresses the implications of the policy trend that encourages data sharing or at least expects it.

## CONCLUSIONS

The present study identified the significant factors that influence the strength of a journal's data sharing policy, specifically the probability of having no data sharing policy as well as that of having a strong data sharing policy compared to a weak policy. Subject area was a commonly identified factor, indicating that life science journals were more likely to have either a strong or a weak policy in comparison to physical science journals and health science journals, respectively. Journal impact factor was positively associated with the likelihood of having a strong policy relative to a weak policy.

It is interesting to note that non-commercial publishers were more likely to have no data sharing policy in comparison to commercial publishers, which tended to have at least a weak policy. The result is in contrast to the findings reported in a previous study, which implies a change in the commercial publishers' reactions to data sharing. It is also worth mentioning that the publishers' location might affect the strength of their data sharing policies. Europe has several national initiatives that enhance and require open data. Thus, it is possible that journals by publishers in Europe are influenced by these actions and have stronger data sharing policies. The type of publisher and journal publisher location are factors that have been less frequently recognized than impact factor and subject area. This study contributes to the existing literature by empirically identifying additional factors that are associated with the strength of a data sharing policy

This study has some limitations that should be addressed in future research. First, for this analysis, the strength of the data sharing policies was categorized as no, weak, and strong according to *Piwowar & Chapman (2008)*, while other studies adopted more nuanced approaches (*Stodden, Guo & Ma, 2013*; *Vasilevsky et al., 2017*; *Resnik et al., 2019*). Specifically, a data sharing policy that only required a data availability statement might not be a "strong" policy since previous studies demonstrated that such statements did not necessarily mean data were actually shared (*Federer et al., 2018*; *McDonald et al., 2017*; *Naudet et al., 2018*; *Rowhani-Farid & Barnett, 2016*). Therefore, future research should incorporate varying degrees of strength in relation to data sharing policies that represent subtle differences in a policy's enforceability. Second, other factors can affect the presence or the strength of a data sharing policy, for example the age of the journals or the language used in the journals (*Gherghina & Katsanidou, 2013*; *Crosas et al., 2019*). The effects of more diverse journal characteristics on the strength of a data sharing policy therefore need to be examined in future research.

### Funding
The authors received no funding for this work.

### Competing Interests
Hye_Min Cho and Jae Hwa Chang are employed by a commercial company: Infolumi Co.

## Author Contributions

- Jihyun Kim and Soon Kim conceived and designed the experiments, performed the experiments, analyzed the data, prepared figures and/or tables, authored or reviewed drafts of the paper, and approved the final draft.
- Hye-Min Cho conceived and designed the experiments, performed the experiments, authored or reviewed drafts of the paper, and approved the final draft.
- Jae Hwa Chang performed the experiments, authored or reviewed drafts of the paper, and approved the final draft.
- Soo Young Kim conceived and designed the experiments, analyzed the data, prepared figures and/or tables, authored or reviewed drafts of the paper, and approved the final draft.

## Data Availability

The raw data is available at Figshare: Kim, Jihyun; Kim, Soon; Cho, Hye-Min; Chang, Jae Hwa; Kim, Soo Young (2020): Raw data of data sharing. figshare. Dataset. https://doi.org/10.6084/m9.figshare.12965096.v1.

## Supplemental Information

Supplemental information for this article can be found online at http://dx.doi.org/10.7717/peerj.9924#supplemental-information.

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
