# Peer review of "Data sharing policies of journals in life, health, and physical sciences indexed in Journal Citation Reports"

_PeerJ, doi:10.7717/peerj.9924_

## Round 0.1 · original submission · Major Revisions

Four reviewers have provided comments on your interesting manuscript and their impressions are generally favourable, as are mine. However, there are some issues that need to be addressed. The reviewers have commented more on the context, coding, interpretation, and reporting of results and I have added a number of, mostly statistical, comments further below which you should respond to in the same way as the reviewers’ comments.

Reviewer #1 asks a question about inter-rater reliability that is also related to those raised by Reviewers #3 and #4. Reviewer #2 makes some important points about documenting the data provided, a point also made by Reviewer #4. They also make two suggestions that I think you will find useful for the discussion, namely a potential limitation around the search strategy and positioning the study within the Open Science movement. Reviewer #3 makes a number of comments including about the classification of journals (and the implications for this on statistical analyses, which I think you might have addressed through the “multidisciplinary” category as explained on Lines 105–108, although the first sentence isn’t perhaps as clear as it could be and Lines 169–170 seems clearer to me on this point; see also comments by Reviewer #4) and the coding of data sharing policies (a point also raised by Reviewer #4). The latter leads to the research question being addressed here and I look forward to seeing your response to their interesting suggestions about this. They have provided some references, which you are free to use in your manuscript if you find them helpful. Reviewer #4 has kindly provided an extensive set of annotations and comments to help you improve the readability and context of your manuscript. I strongly agree with their comment about sharing your Stata code to assist with reproducibility. Their point about extending the discussion into future research is also important (related to this, I wondered what happened to the “Conclusions” section in the abstract.) The process by which you go from 178 categories and 4 quartiles (n=712), to n=709, and then to n=700 needs to be clearly explained (are these numbers due to journals being selected multiple time and journals being in JCR but not Scopus, Lines 100–101?) While I’ve highlighted some of the reviewers’ points here, all of their comments warrant point-by-point responses with each needing either change to the manuscript and an explanation of this or a justification for why you have not made changes in response to that particular comment. The same applies to my comments below.

I wondered about other potential predictors, including whether the journal is online only or included a physical edition, the age of the journal, and the “size” of the journal (for example, the number of articles published annually). You might have good reasons for not looking at these journal characteristics, but I think it would be worth discussing any additional characteristics that you have not looked at but that you feel future researchers could or should do (this is clearly related to Reviewer #4’s comment).

While I appreciate why you have used Chi-squared tests, an alternative that I would recommend would be to replace these with univariable (multinomial or ordinal, see below) logistic regression models. This would allow readers to appreciate any attenuation or intensification of effect sizes between the univariable and multivariable models. There is value in comparing groups of journals both without and with adjustment, I think, and this would provide a more detailed set of results for the unadjusted models as well as a smoother transition between the types of models. Adding effect sizes, including in the abstract on Lines 23–30, along with 95% CIs would help the reader to appreciate not just whether there are differences but also whether the differences are interesting or potentially interesting in a practical sense. At the moment, for example, Lines 184–188 are not particularly informative to readers beyond being statements of statistical significance (or not). The differences between unadjusted and adjusted model results, or the lack of such, could be useful to incorporate into the first paragraph of the discussion I think. Note also that tests for trends (linear or higher) could be added to the regression models to make it easier to describe IF quantile results.

While the assumption of proportionality might well not hold here, did you consider ordinal logistic regression (assuming you feel that the outcome item is ordinal in nature of course, but Table 1 indicates a clearly ordinal scale for all five levels to me). Since you are using Stata, gologit2 would make it relatively straightforward to identify evidence against proportionality and to relax this on a variable by variable basis if this was needed/wanted (see the “autofit” option for this downloadable command).

Note that multinomial logistic regression not produce ORs (odds ratios) but rather RRRs (relative risk ratios). Also, the base category should not influence results and you might want to think about whether by choosing the intermediate level, some information could be lost (the additional results can of course also be obtained from lincom commands in Stata). And/or you could report Wald tests for the independent variables, which will not depend on the base category. Also think about the decimal places used for effect sizes and when reporting very small p-values. When looking at the IF quartile, location, and subject area results, while pairwise comparisons are important, this raises the question of multiple comparisons (if you are not making adjustments for this, you need to be clear about this and describe/temper your interpretations accordingly in the Discussion) and you might want to either make some adjustment here and/or report overall/Wald p-values. Note that Wald tests could provide a useful overall result for both independent variables (e.g. are IF quartiles associated with one non-reference level compared to the reference without specifying a particular combination of the quartiles) and for levels of the dependent variable (e.g. are IF quartiles overall or a particular pairwise comparison of these associated with the level of dependent variable without specifying a particular combination of the latter) and might allow you to report results more simply in places. For IF results, linear (and higher order if appropriate) trends could also be a useful way to report the results in the text. In any case, please ensure that the reader can clearly understand each result (in terms of the specific comparison being reported on, and its effect size, precision, and statistical significance) as they work through the Results section.

Did you consider effect modification/interactions? Were any model diagnostics used to check the models? Did you have any rules (e.g. Peduzzi, et al., 1996) for the maximal complexity of the models given the number of observations in each category of the dependent variable?

I’d like to see some citations for the first two sentences in the Introduction (Lines 34–35 and 35–36). I’m not disagreeing with these statements, but I think they need to be supported by literature or otherwise rewritten to make it clear that they are your opinions. Similarly, for Lines 40–41 and/or 41–42, etc. I suggest carefully working through the Introduction and Discussion to see if any other statements are made that might require references (for at least some readers).

I wondered about the journal selection approach. If I understand the algorithm explained on Lines 89–95, you’ve taken the 100th, 75th, 50th, and 25th percentile journals but not the lowest ranked journal which would have provided 5 evenly spaced points in each category. The bottom (almost) quarter of each category seem to be excluded by this approach. For example, the 2017 ranks for biophysics for the selected journals (Lines 92–95) were (of 72 journals), 1st, 16th, 37th, and 55th (not evenly spaced as I was expecting), leaving those ranked 56th–72nd ineligible for inclusion. It seems to me that these “bottom quarter” journals are potentially interesting in terms of identifying trends. Can you justify this algorithm, including in the manuscript? Also, at least for me, the 2017 JCR has 177 categories in the SCIE edition and 57 in the SSCI edition, for a total of 233 categories. I’m assuming you’re using the SCIE edition, but this is 177 categories rather than the 178 stated on Line 89. If I use the 2018 categories (rather than the 2017 categories, Lines 11 and 89) in SCIE, I get 178 journals but Physics of Life Reviews is now 3rd rather than 1st. Am I misunderstanding some part of this and so looking at the wrong data? The year and edition could also be made explicit on Line 164, along with the date that the data was queried (which is already provided as December 2019).

Specific comments:

Lines 22–23: Do you mean “The quartile of the impact factor…” here? Or “impact factor quartile” (Lines 90, 182, 185, and Tables 2–4)?

Line 34: I would suggest caution about words such as “maximized” unless you are confident that this is a true maxima and otherwise would suggest words such as “increased”.

Line 41: I think this reference to “strength” would be easier for readers if you outlined the possible range of policies before this and referenced the approaches available for coding these. At the moment, a reader might wonder exactly what this “strength” refers to (e.g. mandatory versus comprehensive).

Lines 51–54: Are temporal trends also a potential explanation here? (The lack of evidence is for the two studies from the most recent publication year.) This is simply a thought that occurred to me when reading this, and I haven’t looked at the data collection periods for the 2019 publications, but if such policies were initially accepted by “stronger” journals, it would be reasonable for them to “trickle down” to lower impact journals over time and this could happen quite quickly in publishing.

Lines 79–84: This could perhaps be reordered to identify the knowledge gap first and then to state the study aim. I think this would make a smoother transition from the “what is currently known” outlined just above this paragraph.

Lines 100–101: Please give the number of journals here. A supplement listing the journals showing this process (Lines 89–101) would be very useful if possible.

Line 108: You use “multidiscipline” here rather than the more usual “multidisciplinary” (which you use on Lines 106 and 175). This is also the case in Table 2 and on Line 169. Should these uses all be consistent or is this difference intentional?

Lines 146–150: This seems to be ordinal rather than nominal as “mandates data sharing” must be higher than “expects data sharing” (“mandates” trumps “expects”) and “mandates data sharing and peer reviews” must be higher than “mandates data sharing” (by adding peer reviews). Similarly, “encourages data sharing” seems to me to be lower than “expects data sharing”, and “no policy” must be the lowest of all options. Also Line 152. If you feel that this is, however, nominal, can you please explain your reasoning? Table 1 makes it look very much like this is an ordinal item to me.

Line 160: Note, as mentioned above, that multinomial logistic regression produces relative risk ratios (RRRs) and not odds ratios (ORs). (These are the same for binary outcomes but not otherwise.) This also involves other parts of the manuscript (e.g. Lines 199, 201, etc. and the tables).

Lines 197–198: Personally, speaking as a biostatistician, I wouldn’t find this information useful, but if retained, I would suggest using “p<0.001” for very small p-values rather than “p=0.000”. There are other instances of the same p-value (“0.000”) in other places.

Line 201: I’m not sure you can justify the 3rd decimal places for the OR (RRR I believe) and CI limits here.

Line 207: It’s not clear in the text why there are two p-values here and if these are retained, the specific comparisons these are related to need to be made clear. When looking at Table 4, there is a lack of an overall test (e.g. a Wald test) and/or accounting for multiple comparisons (based on the statistical methods, where the overall level of significance and this point should be clearly explained together) and as mentioned above, you could look at Wald tests and/or tests for trend to help explain these results. Related to this on the following line, it’s not clear what the location or subject area p-values relate to on first reading.

Lines 220–221: Note that “tendency” is sometimes used (with some controversy) to refer to p-values not quite statistically significant (e.g., 0.05<p<0.10) and I suggest a different word choice here. This might be getting into a discussion point anyway.

Line 238: I’d suggest avoiding causal language such as “influencing” unless you are certain that these are actual causes and not just associations (as you phrase it on Line 241, for example) which would require arguing that there are no unmodelled confounders at least. See also Line 296 (same word), Line 254 (“[t]he effect”), and Lines 276 and 287 (“affected” and “affecting” respectively), and perhaps elsewhere.

Line 275: The reference to established norms and practices here doesn’t seem to be part of this study per se.

Lines 291–293: Again, this seems more speculative than is worded here (to me).

Table 2: You could show the combined number and percentage for “Strong policy” as well as its three components. The quartiles are based, as I understand it, on the top journal in each from each category and this interpretation is not clear in the table (as well as the text).

Table 3: As mentioned, I think using logistic regression here would be useful in providing effect sizes rather than four highly statistically significant results overall, which the reader then needs to try to interpret themselves in terms of what differs and its practical significance. Again, the quartiles could be more clearly described.

Table 4: The addition of Wald tests (both for levels of the dependent variable and for independent variables) would help readers to appreciate the degree of evidence here, I think. The choice of base categories for the dependent variable is arbitrary (Lines 196–197) but I think the use of the middle category could be challenged (it omits the most extreme comparison of no policy again a strong policy, which you could also extract using lincom statements in Stata using your models). Again, please consider the decimal places for the effects and CI limits here and the use of “<0.001” for very small p-values. Finally, see above comments on explaining the meaning of the quartiles in the table.

Data: It’s up to you whether you want to include the “result” tabs in the spreadsheet, but note the spelling of “Publihser” for one of these tabs.

·

Basic reporting

The article is well-written, clear, and well-organized.
Tables are all clear and useful.
Background literature is reviewed and discussed; key sources are cited.

Experimental design

Hypotheses are stated and tested.
Methods, including statistical methods, are well-described and appropriate.
Data are relevant to the hypotheses.
Data are shared.

Validity of the findings

The conclusions are well-stated and supported by the data.
The interpretations are sound.
The results are interesting, important, and make valuable contribution to the literature.

Additional comments

I enjoyed reading this article and look forward to its publication.
Excellent work!
One minor point: how often did the researchers who categorized the data disagree? Was inter-rater agreement assessed statistically?

·

Basic reporting

The writing is clear and easy to read. The references to the literature are sufficient and well-referenced.

Experimental design

The experimental questions are clear and well defined, and design is well done. The culture around data sharing and open science is evolving, so it is relevant to re-analyze data journal sharing policies and update the analyses of studies that have previously been done.

Validity of the findings

The results are robust and well-reported. It is great that the authors shared their raw data with the submission, which promotes transparency and reproducibility of the study. I recommend the authors share their data in an open, citeable repository, and cite the code/data per FORCE11 Data Citation Principles, Software Citation Principles (https://www.force11.org/software-citation-principles). The data could be shared in GitHub and published in Zenodo.

Regarding the shared dataset, there needs to be a data dictionary or accompanying ReadMe file, to ensure the data is interpretable and reusable to downstream users.

Additional comments

Sometimes data sharing policies may be more nuanced and just searching on the keywords may cause some of the policies to be missed. This should be noted in the discussion.

Line 221 - Life science journal should be plural

The discussion could be expanded to include more discussion of the culture of open science, and some of the motivations for sharing data.

·

Basic reporting

Generally the papers is well written and referenced. The tables and figures are understandable and appropriate.

I find the discussion of the journal subject areas starting on line 169 confusing. I had to read this several times and I’m still not fully sure I understand how everything was divided up. I think a table or figure might more clearly explain how many journals are in each category and where there is overlap. Table 2 partly does this, but it doesn’t totally make clear what exactly is in the “general” category. I also get the sense there’s some overlap between some of the different journal categories and I can’t figure out whether journals that had more than one category (e.g. were those that were both life science and health science just counted once as general, or counted once for life science and once for health science?).

Experimental design

1. I have some questions about collapsing the various policies levels into just three categories (none, weak, or strong). I appreciate the reference to Piwowar and Chapman, but they’re measuring something slightly different, which is just whether the policy would be enforceable or not. That paper is over ten years old and I see the data sharing landscape to be quite different since that time, so I think an update to that analysis would be helpful. I could potentially see grouping policies with a 3 or 4 as being strong, but those that are a 2 (just requires a data availability statement) to me would be at best a moderate policy, since there are several studies that have demonstrated that just having a data availability statement often does not mean data are ever actually shared (Federer 2018, McDonald 2017, Naudet 2018 and Rowhani-Farid 2016). So I would suggest that policies that only require data availability statements would have a only moderate impact on data sharing and I think doing the analysis based on categories (none, weak, moderate, and strong) could provide more meaningful results.
2. In discussion of the pair coding process, I would like to see more detail about the extent to which their were disparities among coders. Ideally I would like to see a quantitative measure of interrater reliability like the kappa statistic (see for example McHugh 2012). I would also like to hear more about the consensus-building process for disparities. If there were discrepancies, how did the group decide which code to pick? This is coding process is a really important point because the entire analysis relies on the strength of policy category and it bears more discussion here. I think there’s even a little ambiguity with the words that the authors describe indicating obligation – “must” and “require” are pretty clear but I think it’s debatable whether “should” would be expecting or encouraging data sharing. I would imagine there are other types of language that could be similarly ambiguous.
3. If journals with more than one category are counted once for each category, there needs to be some discussion about how that was handled in the statistical analysis.

(i have included full citations to the articles referenced here in the general comments below)

Validity of the findings

All of the findings are appropriately supported and I appreciate the inclusion of the underlying data.

Additional comments

This paper adds to the existing literature in the area by providing more current results (especially important in light of rapidly evolving practices in this area). The analyses were well designed and implemented, though I think some additional description of the methods could be helpful as described above. My main question is just whether the definition of strength of data sharing policy used here answers a meaningful research question. Is the question simply whether the journal has an enforceable policy that has something to do with data? That is answered here, but to me the more interesting question would be whether the journal has an enforceable policy specific to data sharing, since, as discussed above, having a data availability statement often does not mean data are indeed shared. A more interesting question to me is, as the authors state on line 84, “how stringent their policies are regarding data sharing.” I’m really curious to know more about the distinction between journals that just require a data availability statement (which in plenty of cases turns out to be something to the effect of “we can’t/won’t share this data”) versus journals that make a strong stand in saying “you must share your data if you want us to publish your work.” While this paper is interesting as written, I think this additional analysis would provide far more depth.
Articles referenced in the review
Federer, L., Belter, C. W., Joubert, D. J., Livinski, A., Lu, Y.-L., Snyders, L. N., & Thompson, H. (2018). Data sharing in PLOS ONE: An analysis of Data Availability Statements. PLoS One, 13(5), e0194768. https://doi.org/10.1371/journal.pone.0194768
McDonald, L., Schultze, A., Simpson, A., Graham, S., Wasiak, R., & Ramagopalan, S. V. (2017). A review of data sharing statements in observational studies published in the BMJ: A cross-sectional study. F1000Research, 6(0), 1708. https://doi.org/10.12688/f1000research.12673.1
McHugh, M. L. (2012). Interrater reliability: the kappa statistic. Biochemia Medica, 22(3), 276–282. Retrieved from http://www.ncbi.nlm.nih.gov/pubmed/23092060
Naudet, F., Sakarovitch, C., Janiaud, P., Cristea, I., Fanelli, D., Moher, D., & Ioannidis, J. P. A. (2018). Data sharing and reanalysis of randomized controlled trials in leading biomedical journals with a full data sharing policy: survey of studies published in The BMJ and PLOS Medicine. BMJ (Clinical Research Ed.), 360, k400. https://doi.org/10.1136/BMJ.K400
Rowhani-Farid, A., & Barnett, A. G. (2016). Has open data arrived at the British Medical Journal (BMJ)? An observational study. BMJ Open, 6(10), 1–8. https://doi.org/10.1136/bmjopen-2016-011784

Reviewer 4 ·

Basic reporting

Overall this article is written fairly well though there are areas where the language could be improved for comprehension. Please see annotations to manuscript for suggestions in this regard. There are also several areas of background/context that could be expanded upon in order to strengthen the manuscript.

Specific suggestions:
• Tighten up abstract to better reflect the manuscript – see edits to manuscript for some suggestions
• Lines 37-42 Please cite additional evidence as to how data sharing impacts the value of research data.
• Lines 73-82 Can examples of data sharing requirements outside of Europe and the United States be added? I appreciate that Europe’s are the most robust, but I believe it is worth calling attention to the status of data sharing in other areas of the world.
• I would encourage listing all continents out when reporting on journal publisher location rather than lumping some under “Other”. If authors prefer use of “Other”, then please indicate in Tables, which continents are included in that category. If there are no journal publishers represented the sample from certain continents (e.g., Africa, South America) then I’d encourage a sentence added to indicate that in the manuscript (potentially following Line 149).
• Throughout the manuscript and tables, authors need to be consistent in use of plural or singular forms (e.g., “science” or “sciences”) and spelling of words (i.e., multidisciplinary vs. multidiscipline)

Experimental design

The design of this study is consistent with others performed in this area though additional discussion on the method used for categorizing policies, and the review of the policies themselves, would be beneficial.

Specific suggestions:
• Can the authors provide more explanation as to why this broad categorization method (i.e., no, weak, strong) was chosen, especially when more recent studies categorizing data sharing policies have adopted a more nuanced approach (e.g., Stodden et al. 2013; Vasilevsky et al. 2017; Resnik et al. 2019) to tease out the varied degrees of “strength” in journal data sharing policies?
• I find the “multidisciplinary” subject area to be unclear both in definition and reporting. Furthermore, it is not indicated as a (subject) area of interest in the Introduction (or Title for that matter). Other studies have chosen to exclude this category of journal in order to better compare across subject areas/disciplines (Resnik et al. 2019). Is this something the authors considered? If this subject area of journals remains included, I suggest better definition and indication of which journals are assigned to this category.
• Were attempts made to assess the inter-rater reliability between the pairs when reviewing data sharing policies? In studies like these, a challenge is the subjectivity of data sharing policy language. It would strengthen the methods section for the authors to address in more detail how they accounted for and resolved that subjectivity.
• Why were 9 journals excluded from the review/analysis (i.e., 709 journals identified, but only 700 analyzed)?

Validity of the findings

The findings of this research generally align with the research question and the experiment design. However, changes to the manner in which the manuscript reports the findings would help more effectively communicate the results.

Specific suggestions:
• The reporting of “no policy” vs “weak policy”; “weak policy” vs “strong policy” is challenging to follow at times. Authors should review and consider rephrasing these parts of the manuscript to streamline for clarity.
• More context is needed for the findings reported on policy strength for certain disciplines (i.e., Table 5). While this provides an added layer of insight, reporting of this needs to be better integrated into the manuscript and the language used to describe it needs to be consistent (i.e., decide between use of disciplines vs sub-category vs subject).
• There are (seemingly) anecdotal observations included in the reporting of findings (for example, lines 270-278; 299-301). Added context as to how these findings were determined/collected would strengthen reporting.
• The authors do a nice job contextualizing their findings in the body of existing literature.
• The raw data shared is very helpful, but could benefit from additional descriptive information (e.g., readme, data dictionary) to assist future researchers/readers. I’d also encourage the authors to consider sharing the files associated with their analyses conducted using Stata in order to ensure full reproducibility of the study. Finally, I recommend authors deposit all raw data and analyses files in an appropriate repository (examples here: https://peerj.com/about/author-instructions/#data-and-materials).

Additional comments

This is an interesting study that contributes to knowledge around journal data sharing policies in the areas of life, physical, and health sciences. There are times where the manuscript’s language is a bit challenging to follow – especially in the reporting of results – however, this study provides added insight on factors that may influence the presence and strength of journal data sharing policies. In addition to the comments made, I strongly recommend that the authors flush out the Conclusion to include elements such as: what are next steps? Are there areas of this research that could benefit from further study? How might the limitations acknowledged by the authors in this study be addressed moving forward?

Annotated reviews are not available for download in order to protect the identity of reviewers who chose to remain anonymous.

---

## Round 0.2 · Minor Revisions

Thank you for your thorough responses and constructive revisions. Thank you also for your patience during this challenging time. I think your interesting and important manuscript was improved by your revisions in response to the reviewers very helpful comments.

Reviewers #3 and #4 have both raised a small point regarding the revised manuscript that you need to address about NHI policy. There are two additional comments from Reviewer #4 that you might want to take on board this stage. They have asked whether additional documentation could be provided for the data and Stata code. If possible, this would be a useful addition, particularly labelling the values of the data (see my comment, and some code, below). They have also made a valid point about the ending of the manuscript, which you might also want to take advantage of here. I’ll ask you to address these minor points, along with some additional comments I’ve made below. Once you have made changes or explained why you do not feel the need to make changes for each of their and my comments, and assuming no new issues arise, I expect to be able to accept your manuscript.

Thank you for adding your data on figshare. Could you please add a mention of the availability of files here somewhere in your manuscript? I had asked “about sharing your Stata code to assist with reproducibility” and while there is a file there named “Stata code_200411.dta” this is a Stata data (.dta) file rather than a Stata code/syntax (.do) file. A similar point arises with the supplementary file labelled “statacode” called “Stata_code_200411.xlsx”, which is also data and could be replaced by the Stata .dta file on figshare. If you are able to add your .do file to figshare and/or as a supplement to your manuscript, that would be extremely helpful to readers wishing to reproduce your analyses. Related to a comment from one of the reviewers, could you add variable labels to your .dta file? This could be as simple as the data file after running the below code (if I have understood your coding correctly), or you could add this code to your Stata .do file:

label define policy 1 "No policy" 2 "Weak policy" 3 "Strong policy"
label values policy policy
label define publisher 1 "Commercial" 2 "Non-commercial"
label values publisher publisher
label define location 1 "North America" 2 "Europe" 3 "Others"
label values location location
label define subject 1 "Life science" 2 "Health science" 3 "Physical science" 4 "Multidiscipline"
label values subject subject
label define IF_quartile 1 "Q1" 2 "Q2" 3 "Q3" 4 "Q4"
label values IF_quartile IF_quartile

While not impossible, it was striking that the exact same point estimates and 95% CIs occur for two analyses “(RRR, 2.99; 95% CI, 1.85 to 4.81).” (see Lines 37 and 40). This point led me to check some, but not all, of your analyses and I’m not sure that all of your reporting is correct, although I could be misunderstanding some of the models. For “policy_level”, 1=none, 2=weak, and 3=strong, your multivariable (see comment below about nomenclature) analyses for the result on Line 33 (RRR for strong versus weak policy of 0.36 for life science versus physical sciences) seem like it should be based on something like: “mlogit policy_level i.IF_quartile i.location i.publisher b3.subject, base(2) rrr”, which would give a RRR of 2.77. I’m assuming that the 0.36 here is the reciprocal of this, but the wording is “were more likely” and so I expected to see an RRR > 1. If this query is correct, it would be worth checking all other values just in case other such issues have slipped into the manuscript.

Slightly related to this, I wasn’t sure if the “more likely to have a data sharing policy” on Line 36 meant more like to have weak compared to none or weak/strong combined compared to none and wasn’t able to replicate the RRR of 2.99 on Line 37 through some quick experiments (e.g., RRR for weak versus none would be 7.87, as reported on Line 311 and in Table 6), but as noted above this could be a typo where the value from Line 40 was used here. In any case, you should check the manuscript for any other slightly ambiguous phrasing as well.

While your analysis treats the dependent variable as nominal, I still think that the three categories are clearly ordinal (Line 186). The analysis is the same as if this was nominal, but none is clearly < weak which is clearly < strong, so the measure itself is ordinal (as mentioned previously). Again, on Line 191, the measure was ordinal rather nominal, but it was the failure of the model to satisfy proportionality (Line 194) that led to mlogit for the analysis rather than ologit. mlogit still deals with ordinal dependent variables, it simply does not assume a shared effect for lower versus higher levels (i.e., the assumption of proportional odds).

While I’m being slightly pedantic here, “quartile” technically refers to the cut-points rather than the quarters themselves. (See https://en.wikipedia.org/wiki/Quartile.) So, for example, on Line 117, rather than “the highest-ranked journal in each journal impact factor quartile”, you could replace “quartile” with “quarter” to be more precise and explain that Q1 is 0.25, Q2 is 0.5, etc. on Lines 117–118). I appreciate that usage of this term has drifted away from this technical meaning so if you want to keep the present wording here and throughout the manuscript, that’s also fine.

As another pedantic comment, “univariate” is often used for a single dependent variable, with “multivariate” for hierarchical data. This leaves “univariable” for a single independent variable and “multivariable” for multiple regression. This is discussed in http://doi.org/10.2105/AJPH.2012.300897 if you are interested. If you agree, this would change Lines 24, 26, 189, 195, 197, 226, 228, 233, 239, 240, 248, 255, 256, 284, 308, 310, 312, 313, 316, and possibly elsewhere, and the titles for Tables 3, 4, 6, and 7. All of your models are univariate, and they are either unadjusted (univariable) or adjusted (multivariable) using this nomenclature.

When you say (Line 240), “All possible interactions for the variables…” do you mean “All possible two-way interactions for the variables…” or did you also look at higher order interactions? This should be noted back in the statistical methods so the reader is prepared to see tests of interactions in the results section. See below for a related comment about the use of AIC/BIC.

I think that all of the method should be described in that section rather than in the results, so I feel that the checks on overfitting (Lines 249–253) should be covered in the methods (that this was going to be done and the criterion/criteria for being acceptable, including both references) leaving only the actual EPV needing to be covered in the results.

For the tables, you could consider reordering these so that the current Table 2 (defining the policies) goes before the current Table 1 (which uses the policy categories in Table 2).

In Table 3, I think you could delete the beta and SE columns as these are only useful insofar as they allow for the calculation of the RRR and its 95% CI (which are already included). If you did this, I’d move the p-value columns to after their associated RRR and 95% CI columns (i.e., RRR, 95% CI, p-value). With this additional space, Table 4 could then be incorporated into Table 3 as extra columns (I’d be happy with just the p-value column being added, so a single additional column there could replace this table and should fit with the beta and SE columns removed). The “(all)” rows, Wald tests for all equations involving that variable, would then be in the same rows as in Table 4 for the variable name. The same would apply for Table 7, which could be a column (for p-values) added to Table 6, which could itself have fewer columns. This change isn’t required, but I think that it would simplify the tables and better connect related pieces of information for the reader, perhaps in exchange for losing the Chi-squared statistics (which I can’t imagine readers being interested in) and degrees of freedom (which they could calculate themselves, but again, I can’t imagine readers being interested). Alternatively, you are welcome to keep the Chi-statistics and degrees of freedom for any arrangements of the tables.

Table 5 shows lower AIC for the model with the interaction and lower BIC without, which is one of the issues with looking at multiple ICs. There are specific arguments for using AIC or BIC (or another IC) in any particular situation, but at the moment, I can’t see how you have justified using BIC to make the decision about which model is “better” and for a reader who would themselves use AIC, the decision would be reversed. Note also that you use commas separating the thousands for the main effects model but not for the interaction model and this should be consistent (here and throughout the manuscript). I suggest adding some text to the statistical methods explaining which ICs were investigated, or that only BIC was investigated if AIC was not going to be incorporated into the decision-making process, and what the decision rule for selecting a model as “better” was. This would go with the additional methods text around the interactions I’ve requested above, including the reference on Line 247.

For Figure 1, I suggest you replace these figures with ones without shading. You could use “set scheme s1color” or “set scheme s1mono” to change the graph scheme to one without the background, set the graph elements individually, or use the graph editor to achieve this. I also suggest a space before the opening parenthesis on the three y-axes. You could also consider having the three graphs in a single column (option “col(1)” if you are using “graph combine”), to avoid the “hole” at the bottom right, but these points about shading and panel arrangement are clearly just a matter of taste.

·

Basic reporting

The reporting is clear and the edits/additions the authors have made have improved the clarity and thoroughness of this reporting.

Experimental design

I appreciate the added statistical testing the authors conducted.

Validity of the findings

The authors provide a useful conclusion that also points to some new areas for future research.

Additional comments

A very minor point of clarification: lines 95-96 say that NIH requires a data management and sharing plan. That is not the case at the moment – is has not yet been determined when that policy will go into effect.

Reviewer 4 ·

Basic reporting

The authors have done an excellent job incorporating the feedback from reviewers, which has greatly improved the clarity and communication of the impact of this research. After reading through the revised manuscript, I have just one point where additional clarification is needed:

Lines 94-96 – the National Institutes of Health (NIH) currently does not require data management and sharing plans broadly. The citation referenced in the manuscript is to a draft of a potential policy that the NIH may adopt. My suggestion is to reword that sentence to read something like, “For example, the National Institutes of Health indicates they may expand upon existing data sharing requirements and require data management and sharing plans more broadly moving forward (National Institutes of Health, 2019).”

Here is a link to the existing NIH data sharing requirements for context: https://grants.nih.gov/policy/sharing.htm

Experimental design

Authors addressed all comments made on the original manuscript, which has resulted in better description of the research design and process. I have no further comments.

Validity of the findings

Again, the authors sufficiently clarified the findings of this research by providing additional explanation or adding citations. Thank you as well to the authors for uploading their raw data and Stata code to Figshare. A suggestion I would make is to consider adding documentation to these files to help contextualize the process of analysis and better assist potential future users of the data.

Additional comments

The authors have revised this manuscript and it now more strongly communicates the outcomes of their research. A final comment I will make is in regards to the Conclusion. While the authors did an excellent job of summarizing limitations and future research, I still feel the paper ends rather abruptly. I would encourage adding some additional lines that perhaps situate this manuscript in the context of existing literature, reiterate how it contributes to that literature, and how it provides a stepping stone for further research in this area moving forward.

---

## Round 0.3 · accepted · Accept

Thank you for your revisions and responses. I am delighted to accept your manuscript and look forward to seeing it stimulate more discussion and research in this area. The sole reviewer has made a single copyediting comment, and I’ll make a few myself below, for you to address in preparing the final version of your manuscript.

Line 46: I wonder if you might consider a “Conclusions.” subheading in the abstract about here (before “Future…”) to separate this from the results. You could also move some of the interpretative statements from the results section (Line 39 [although “Contrary to some existing research,” could simply be removed instead], Lines 41–42, and Lines 45-46) into these “Conclusions”, e.g. “These findings may account for the increase in commercial publishers’ engagement in data sharing and indicate that European national initiatives that encourage and mandate data sharing may influence the presence of a strong policy in the associated journals. Future research needs to explore the factors associated with varied degrees in the strength of a data sharing policy as well as more diverse characteristics of journals related to the policy strength.” A final option might be to make the subheading on Line 33 “Results and Conclusions.”

Line 112: “affected by” would be causal language and I suggest “associated with” would be more reasonable for observational data. This would then match the use of “related” on Line 114.

Line 182: More usual would be “…from THE Netherlands…” (see https://en.wikipedia.org/wiki/Netherlands).

Line 183: Oceania seems to be usually defined as including Australia (see https://en.wikipedia.org/wiki/Oceania) so perhaps “…in Asia and Oceania (including Australia).”

Lines 183–184: I’m not sure what “None in the ‘others’ category represented the sample from the continents.” here means. I think it can safely be deleted without loss of information.

Line 188: As you note below on Line 193, this variable has ordinal characteristics (the three levels can be ordered, making it ordinal), so I suggest simply deleting “nominal” from here.

Lines 192–193: The comma between “conducted” and “because” here isn’t needed but I don’t think any of “because the dependent variable—the strength of the data sharing policies—was measured at the nominal level.” is in fact needed (and conflates the measurement level of the dependent variable [ordinal] with how it was modelled [nominal]). I think removing this would also improve the flow: “Both a univariable and a multivariable multinomial logistic regression analysis were conducted. Although the dependent variable had ordinal characteristics, we used multinomial rather than ordinal logistic regression because…”

Line 317: Perhaps “…the effect SIZE was…” (as you use on Line 322 in the same context).

Line 343: Similar to Line 112 above, “association with” rather than “effect on” here would avoid appearing to claim causality from observational data.

Line 368: Similar to Lines 112 and 343 above, “was associated with” rather than “affected”.

Line 400: “Journal impact factor WAS positively ASSOCIATED WITH [not “affected”] the likelihood…”

References: DOIs are provided inconsistently (e.g. for the second reference but not the first, both of which are from the same journal, and is missing for a PeerJ article later). If included for some references, these should be provided for all references where these are available. The DOIs are sometimes formatted inconsistently (note the use of underlining for some but not all DOIs). Similarly, some but not all URLs are underlined.

Figure 1: If you can improve the aesthetics of this figure by removing the shading and considering using either a single row or a single column (to avoid the “hole” at the bottom right), that would be very useful.

Reviewer 4 ·

Basic reporting

The authors have incorporated all suggested changes satisfactorily. Quick copyediting note - in the reference list, the National Institutes of Health (NIH) citation needs an "s" added to "Institute".

Experimental design

The authors have incorporated all suggested changes, which has resulted in increased clarity in their results reporting.

Validity of the findings

No further suggestions. Thank you to the authors for strengthening the documentation and description for the code shared in Figshare.

Additional comments

I have no further suggestions or comments on the manuscript at this time.